# Non-Coding RNAs in Airway Diseases: A Brief Overview of Recent Data

**DOI:** 10.3390/cancers15010054

**Published:** 2022-12-22

**Authors:** Giusy Daniela Albano, Rosalia Gagliardo, Angela Marina Montalbano, Mirella Profita

**Affiliations:** Istituto di Farmacologia Traslazionale, Consiglio Nazionale delle Ricerche, Italy (IFT-CNR), 90146 Palermo, Italy

**Keywords:** microRNAs, asthma, COPD, lung cancer, pollutions, extracellular vesicles

## Abstract

**Simple Summary:**

Nc-RNA are microRNA, long-coding RNA, and circulating-RNA. In this review we report most recent data regarding the role of nc-RNA in airway diseases, with a particular attention to microRNA. They are short, endogenously initiated non-coding RNAs involved in post-transcriptionally control gene expression via either translational repression or mRNA degradation. MiRNAs play significant roles in control of cell mechanisms involved in developmental timing and host-pathogen interactions as well as cell differentiation, proliferation, apoptosis, and tumorigenesis. Today the knowledge of the functions of the micro-RNA are of fundamental importance to define the subtypes of inflammatory diseases of the lung and to understand the effectiveness of the treatment.

**Abstract:**

Inflammation of the human lung is mediated in response to different stimuli (e.g., physical, radioactive, infective, pro-allergenic, or toxic) such as cigarette smoke and environmental pollutants. These stimuli often promote an increase in different inflammatory activities in the airways, manifesting themselves as chronic diseases (e.g., allergic airway diseases, asthma chronic bronchitis/chronic obstructive pulmonary disease, or even lung cancer). Non-coding RNA (ncRNAs) are single-stranded RNA molecules of few nucleotides that regulate the gene expression involved in many cellular processes. ncRNA are molecules typically involved in the reduction of translation and stability of the genes of mRNAs s. They regulate many biological aspects such as cellular growth, proliferation, differentiation, regulation of cell cycle, aging, apoptosis, metabolism, and neuronal patterning, and influence a wide range of biologic processes essential for the maintenance of cellular homeostasis. The relevance of ncRNAs in the pathogenetic mechanisms of respiratory diseases has been widely established and in the last decade many papers were published. However, once their importance is established in pathogenetic mechanisms, it becomes important to further deepen the research in this direction. In this review we describe several of most recent knowledge concerning ncRNA (overall miRNAs) expression and activities in the lung.

## 1. Lung Diseases

Lung diseases are types of disorders affecting the normal pulmonary functions preventing the ability to breathe. Many respiratory diseases begin with the onset of inflammatory reactions with a key role in the pathological conditions of the lung, as the reduced airflow of asthmatic and Chronic Obstructive Pulmonary Disease (COPD) patients [1].

The causes of the asthma pathogenesis are genetic, immune, or associated with environmental factors (pollutants, allergens, and pathogens). Asthma is characterized by airway hyper-responsiveness (AHR) and leads to intermittent and usually reversible airway obstruction. The symptoms of asthma are generated by drugs, exercise, or intrinsic problems; they can be repeated wheezing, chest tightness, cough, and other symptoms all related with reversible airflow restriction. Asthma pathogenesis is regulated by the co-participation of various immune cells and cytokines that are highly heterogeneous and affect numerous aspects of asthma regarding pathologic processes and clinical manifestations. Airway inflammation is characterized by T-helper 2 cells (Th2 cells) immune response and type 2 cytokines (e.g., IL-4, IL-5 and IL-13) release, eliciting airway eosinophilia, bronchial hyperresponsiveness, mucus overproduction, airway remodeling, and immunoglobulins E (IgE) synthesis in asthma. In addition, subject with asthma show in the airway epithelial cell exfoliation, fibrosis, vasodilation of the walls, and exudation of plasma [2]. The World Health Organization described that people suffering from asthma are currently 235 million, worldwide, and children are a bigger and most vulnerable group of asthmatic population [3]. Pharmacological treatment of asthma includes its action against allergens and irritants, as well as the use of β-adrenergic receptor agonists such as bronchodilators, glucocorticoids, and antileukotrienes to suppress inflammation or immunotherapy. These conventional therapies of asthma patients keep under control the symptoms but are not able to cure completely the disease [4].

Cigarette smokers are approximately 90% of the COPD patients. Smoke habits are the first cause of risk factor for the development of COPD [5,6]. The inflammation in COPD involves the innate and adaptive immune responses. It is one of the major public health problems worldwide since it is the major cause of chronic morbidity and mortality. The subjects with COPD are older than forty years of age and had a progressive and irreversible airway obstruction, characterized by deterioration of pulmonary function time related [7], causing the destruction of the lung parenchyma. This characterizes inflammation of central and distal airways, pulmonary emphysema, respiratory bronchiolitis representing the hallmarks of COPD [5,6]. The inflammation in COPD subjects includes the recruitment of different cell types as neutrophils, macrophages, lymphocytes, and the activation of epithelial cells in the airways [5,6]. To what extent central airways may mirror events, occurring in distal lung is uncertain. β_2_-AR-agonist and anticholinergic drugs are as concurrent therapy used in the treatment of COPD to maximize bronchodilation and have a potential anti-inflammatory role [8]. The older compounds are modified to make novel therapeutic agents more potent, long lasting, enclosed in new inhalation devices [9].

Asthma and COPD are diseases with similarities [10]. However, they are different regarding etiology, type of inflammatory cells, mediators, consequences of inflammation, response to therapy [8] (Figure 1). COPD can originate in smoker subjects aged over 40 and its incidence increases in over 60-year-olds. Aging accelerates the decline of normal lung function in COPD patients showing a premature lung function loss.

The mechanisms of aging can be non-programmed or programmed. The first is related to the failure of the repair of DNA in the organs due to the increase in oxidative stress. The second is related to telomere shortening for a repeated cell division. Both aging defects are present in COPD patients per se, as physiological conditions disease. They may be involved in the progression of diseases toward lung cancer (LC). The premature aging of the lung, genetic predispositions, common pathogenic factors as growth factors, activation of intracellular pathways, or epigenetic modifications are common mechanisms that can be a cause of high prevalence of LC in patients with COPD [11].

The major risk factor for lung cancer is cigarette smoking. The smoking habit, common to COPD and LC alters the defense mechanisms regarding the antioxidant production such as superoxide dismutase, anti-proteases, and DNA repair mechanisms. If the damage to these mechanisms becomes too high in COPD patients, mutations leading to LC can occur [12]. In LC, increasing oxidative stress, reduction of DNA repair mechanisms and the resulting DNA damage, chronic exposure to pro-inflammatory cytokines, and increased cellular proliferation are common to COPD. LC is also a leading cause of morbidity and mortality in patients with COPD [13]. Mutations in oncogenes lead to an abnormal cell proliferation causing LC. The benign tumor was transformed to an invasive cancer by additional mutations that initiate spread invasiveness and anaplasia of the cells. It usually originates from the basal epithelial cells and are classified into two types, non-small-cell lung cancer (NSCLC) and small-cell lung cancer (SCLC) [14].

Understanding the cellular and molecular processes driving the modifications in primary cells from patients, together with original and new models of disease is essential for the development of new treatments in lung diseases. Today, knowledge on immunobiology of inflammatory and structural cells of the lung is insufficient to explain the progression of lung disease to cancer [15].

## 2. Nc-RNA Biogenesis

The nc-RNA are classified in two categories housekeeping and regulatory. The regulatory included two noteworthy ncRNA: miRNAs or miRs (transcripts of 19–25 nucleotides) and long-ncRNAs (transcripts of 200 nucleotides) (lnc-RNAs) and circular RNAs (circ-RNAs). MiRNAs generally regulate post-transcriptional gene expression in physiological and pathological processes targeting the messenger RNA (mRNA) cleavage and degradation, and/or by inducing mechanism of translational repression or mRNA degradation [16]. lncRNAs, interacting with DNA, RNA, or protein through various mechanisms, positively or negatively control the stage of gene expression [17,18,19]. Finally, circ-RNAs have multiple biological functions as miRNA sponge, transcription regulator, protein translation, interaction with protein, RNA maturation, and so on [20]. The biogenesis, characteristics, types, and mechanism of action ncRNAs have a relevant role in developmental and homeostasis of human tissues [17,18,19,21]. In this section we described the biogenesis of miRNA mainly reported as nc-RNA in airway diseases in this review.

A conserved family of small endogenous noncoding RNA (nc-RNA) molecules (18–22 nucleotides in length) named microRNAs (miRNAs) was found in eukaryotes and have been extensively studied in human diseases and in cancer [22,23,24]. Many human diseases show aberrant expression of miRNAs [25,26]. MiRNAs are critical for normal animal development and are involved in a variety of biological processes such as proliferation, apoptosis, differentiation, and survival, playing an important role in gene expression regulation [27,28], also in the biological processes of the lung diseases [29].

MiRNA family are long transcripts called clusters, with repeated similar regions [30]. The discovery of new miRNAs and the description of their role in the pathogenesis of diseases are constantly evolving [31,32]. MiRNA action often suppresses expression of target gene interacting with the 3′ UTR of target mRNAs [33], or it has also been reported that miRNAs interact with 5′ UTR, coding sequence, and gene promoters. They shuttled between different subcellular compartments to control the rate of translation transcription [34].

MiRNA biogenesis is classified as canonical and non-canonical. However, the dominant pathway is the canonical miRNAs processing. The biogenesis of miRNAs starts in the nucleus with the processing of RNA polymerase II/III transcripts post- or co-transcriptionally. Pol II initiates the transcription of miRNA genes to produce pri-miRNA. Then, DiGeorge Syndrome Critical Region 8 (DGCR8), a microprocessor complex with a RNA binding protein ribonuclease III, transforms the pri-miRNA in pre-miRNA. Subsequently, DGCR8 targets N6-methyladenylated GGAC and other motifs into pri-miRNA, Drosha cleaves the pri-miRNA duplex in the hairpin structure of pri-miRNA [35]. Subsequently, by the exportin 5 (XPO5)/RanGTP complex action, pre-miRNA translocated from the nucleus to the cytoplasm and was transformed in mature miRNA/miRNA duplex by RNase III endonuclease Dicer and the RNA-binding protein TRBP the action. Then helicases separate the duplex by Argonaute (AGO2), and the strand guide of miRNA incorporates RISC to form a complex that recognizes specific mRNA by sequence complementarity leading to either mRNA degradation or translational inhibition [36,37] (Figure 2).

The multiple non-canonical pathways of miRNA biogenesis involved different combinations of proteins used in the canonical pathway as Drosha, Dicer, exportin 5, and AGO2. miRNA biogenesis is grouped into two principal pathways defined Drosha/DGCR8-independent and Dicer-independent. Drosha/DGCR8-independent pathway produces pre-miRNAs like Dicer substrates. These pre-miRNA such as mirtrons nascent through exportin 1 are exported to the cytoplasm without the need for Drosha cleavage [35].

## 3. Nc-RNAs in Asthma

The throughput epigenetic analysis technologies, together with traditional biological function and clinical studies improved the genetic knowledge and finding epigenetic biomarkers of asthma provided new frontiers in the precision medicine of the lung [38]. The knowledge of nc-RNA regulatory networks opens new perspectives for the understanding the pathogenesis of asthma [39]. Asthma is a heterogeneous chronic inflammatory disorder in which different endotypes contribute to defining clinical inflammatory phenotypes. It is classified as mild, moderate, and severe. Nc-RNAs, in particular microRNA profiling in asthma is involved in the definition of subtyping asthma as potential biomarkers and therapeutic targets [40]. Altered expression of nc-RNAs in blood, in exhaled breath condensate, or in induced sputum condensate of sputum indicate the progression of asthma and the immune response in the lung.

Nc-RNAs regulate the gene expression at the post-transcriptional level and by targeting mRNAs affect the synthesis of cytokines and signaling pathways in airway inflammation [41,42]. mRNAs induce the activation of structural cells (bronchial epithelial cells, fibroblasts, endothelial cells, and smooth muscle cells) and immune cells playing an essential role in cell proliferation, differentiation, signal transduction, stress response, cell apoptosis, and other cellular and molecular aspects of asthma diseases.

MiRNAs might be considered as novel biomarkers of disease [42]. For examples miR-21 and miR-155 are important regulators of gene expression of many immunological molecules. Higher levels of miR-21 and miR-155 are detected in the serum of asthmatic patients compared to control subjects. Both miRNAs might be considered as potential non-invasive biomarkers useful for the diagnosis and response to the therapy in eosinophilic asthma [43].

The inflammatory processes of asthma are regulated by the activation and differentiation of Th2 cells, secretion of cytokines, and functions of eosinophils. let-7 family, miR-193b, miR-375 (downregulated), and miR-21, miR-223, miR-146a, miR-142-5p, miR-142-3p, miR-146b and miR-155 (upregulated) represent a core set of nc-RNA involved in asthma. Many of them are involved in T-cell differentiation increasing Th2 cell phenotype and Th2 cytokines secretions active in the origin of hyperplasia and hypertrophy of bronchial smooth muscle cells [44,45,46]. miR-21 is involved in the switch of Th1 versus Th2 responses, and defines the mechanisms of immunoinflammatory responses, limiting in vivo immune response-mediated activation of the IL-12/IFN-gamma pathway [47]. miR-146a is a candidate molecule with an association with impact of genetic variation in asthma [48]. It together with miR-26a and miR-31 is increased in the lung tissues of asthma mice, and in bronco alveolar lining fluid (BALF) of asthma children [49]. Furthermore, miR-146a define endotypes of asthma in moderate asthma (MA) and severe asthma (SA). MiR-146a and lower production of resolvin D1 create a dysregulation of inflammation in children, promoting remodeling processes and leading to lung function impairment [50]. MiR-155 and miR-221 are associated with Th2 responses [51] and with cells involved in allergic response (eosinophils, macrophages mast cells) in asthma and rhinitis [39,52,53,54].

Follicular helper (Th) and regulatory T (Treg) cells are involved in allergic asthma [55]. MiR-17 affects T-cell-like characteristics and via the de-repression of genes encoding effector cytokines transform them in Treg cells. It modulates regulatory T-cell function through targeting eosinophils and by targeting co-regulators of the Foxp3 transcription factor Foxp3 co-regulators [56]. PU.1 transcription factor is a negative regulator of Th2 cytokine release. It is upregulated in the airways of allergen-challenged miR-155 knockout mice. These data underline that miR-155 regulates Th2 responses in allergic airway inflammation by transcription factor PU.1 [53]. Furthermore, miR-155 regulates type 2 innate lymphoid cells ILC2s and IL-33 signaling in allergic airway inflammation [57].

Remodeling and oxidative stress in asthma are regulated by many miRNAs. MiR-26, −133a, −140, −206, and −221, are associated with an effect on smooth muscle cell function and proliferation [39]. MiR-143-3p inhibits airway remodeling in asthma, suppressing transforming growth factor (TGF)-β1-induced cell proliferation and protein deposition of extracellular matrix (ECM) production proliferation via negative regulation of nuclear factor of activated T cells 1 (NFATc1) signaling [58]. MiR-192-5p is down-regulated in asthmatics and attenuates airway remodeling and autophagy in asthma by targeting MMP-16 and ATG7 [59].

The action of miRNAs can favor the progression of asthma phenotype from mild to severe stage [60]. However, despite the relevant role of miRNAs in asthma, few studies define their immunological activity in severe asthma. It is observed that miR-221 downregulates the action of TGF-β, on the aberrant airway smooth muscle proliferation and size, and consequently proinflammatory effects [61]; miR-28–5p and miR-146a/b downregulation led to circulating CD8+ T-cell activation in severe asthma [62]; and miR-223–3p, miR-142–3p, and miR-629–3p are well correlated with neutrophils in severe asthma [63]. All these findings facilitate the conclusions in this field [64] underlining that miRNA expression profiles might represent a risk factor for the development of a severe stage of asthma disease [28].

MiR-1278 inhibited inflammation in asthmatic mice and counteracted the effect of TGF-β1 in the cell proliferation and reduced apoptosis in airway smooth muscle cells (ASMCs). In particular this study showed that miR1278/SHP-1/STAT3 pathway is involved in airway smooth muscle cell proliferation in a model of severe asthma [28]. Recent overviews underline the emerging role of ncRNAs in childhood asthma. For instance, lncRNA CASC2 and BAZ2B are increased in the serum of childhood asthma [65,66]. CASC2 is involved in childhood asthma through inhibiting ASMCs proliferation, migration, and inflammation via miR-31-5p activity [65]. *BAZ2B* correlates with M2 macrophage activation and inflammation in children with asthma, and positively correlates with the exacerbating progression of diseases [66].

Circulating miRNAs such as miR-155-5p and miR-532-5p are predictive of asthma ICS treatment response over time and are significantly associated with changes in dexamethasone-induced trans-repression of NF-κB. Accordingly miR-155-5p and miR-532-5p might be considered as predictive of ICS response in clinical trial [67]. MiRNA-155 and Let-7a are differentially expressed in the plasma asthmatics than in control children, and levels well correlate with the degree of asthma severity. MiRNA-155 and let-7a could be used as serological non-invasive biomarkers for diagnosis of asthma and degree of severity [68]. Furthermore, it is underlined that circulating miR-146b, miR-206, and miR-720 are predictive of clinical exacerbation in asthmatic children, representing diagnostic biomarkers and therapeutic targets in childhood asthma [69] (Figure 3).

All these findings suggest that the regulatory networks of ncRNA provide new tools for the diagnosis and treatment of asthmatic patients to control inflammation, remodeling, and bronchial hyperresponsiveness in asthma controlling the activity of immune cells, ASMCs, and bronchial epithelial cells.

## 4. ncRNAs in COPD

Many studies show that miRNAs production is increased in the pathogenesis of COPD [70]. MiRNAs involved in the myogenesis (proliferation and differentiation of satellite cells) alleviate the negative impact of skeletal muscle dysfunction and mass loss in COPD regardless of the degree of the airway obstruction [71].

miRNA-mRNA regulatory network is identified by GEO2R tool in the circulating plasma of COPD patients. Hub genes are potentially modulated by miR-497-5p, miR-130b-5p, and miR-126-5p and among the top 12 hub genes, MYC and FOXO1 expressions are consistent with that in the GSE56768 dataset [72]. Recent studies describe increased levels of miR-221-3p and miR-92a-3p in the serum of COPD patients than in healthy subjects, suggesting that both miR-221-3p and miR-92a-3p might be considered molecular markers to discriminate stable COPD and COPD with acute exacerbations. In addition, the same authors underline that miR-221-3p and miR-92a-3p are involved in the description of CSE-induced hyperinflammation of COPD [27].

Asthma-COPD overlap syndrome is an inflammatory disease of the airways that describe a new phenotype including both asthma and COPD characteristics. miRNA molecular pathways can help the scientists to better understand the pathophysiological features in many diseases. MiRNA expression profile of serum and sputum supernatants shows the increased expression of miRNA-338 in the sputum supernatants of patients with different obstructive diseases than in peripheral blood, while miRNA-145 increases only in the sputum supernatants of asthmatic subjects. However, both miRNAs are higher in the sputum supernatants of patients with asthma and COPD compared with control subjects. These data describe miRNAs as potential biomarkers in the discrimination of asthma-COPD overlap syndrome, asthma, and COPD [73]. The expression of five miRNAs (miR-148a-3p, miR-15b-5p, miR-223-3p, miR-23a-3p, and miR-26b-5p) is lower in patients with asthma-COPD overlap syndrome. Moreover, these miRNAs might be able to discriminate patients with asthma-COPD overlap and patients with either asthma or COPD. Between these miRNAs, miR-15b-5p is the most accurate and associated with the levels of Periostin and chitinase-3-like protein 1 (YKL-40) in the serum of patients representing a potential marker to identify asthma-COPD overlap patients [74].

The mechanism of cellular senescence is important to drive pathogenesis of COPD. MiRNA-34a is involved in this cell mechanism and reduces sirtuin-1/6 as markers of senescence through PI3K–mTOR signaling. In this manner its activity reduces secretory phenotype associated with senescence, and reverses cell cycle arrest in epithelial cells from peripheral airways of COPD patients [75].

MiRNA data analysis was performed using the TAC software, and reveals 148 miRNAs that are differentially expressed in PBMCs from patients with COPD compared with normal controls. Among the 148 miRNAs, 104 miRNAs are upregulated, and 44 miRNAs are downregulated [76]. The data show that miRNAs differentially expressed might be involved in the regulation of cell processes playing a fundamental role in the pathogenesis of COPD. Accordingly, it is possible to think that future investigation in this direction might provide further insight into the mechanism of COPD.

The miRNAs analyses show increased levels of miRNA-21 in airway epithelium and lung macrophages of the lungs of mice with CS-induced experimental COPD. miRNA-21 inhibitor (Antago-miR21) reduces the miRNA-21 expression in CS-induced lung of mice, suppressing the infiltration of inflammatory cells (macrophages, neutrophils, and lymphocytes). Furthermore, the treatment of with Antago-miR21 CS-induced mice decreases hysteresis, transpulmonary resistance, and tissue damping improving lung function in the mouse models of COPD [77]. Accordingly, it is observed that COPD patients with periodically experience acute exacerbation have increased levels of miR-21 inversely correlated with FEV1. These data support the concept that systemic levels of miR-21 can be involved in the pathogenesis of airway diseases and represent a therapeutic target to control the physiology of the lung [78].

In a little cohort of subjects, classified as COPD, smokers, and non-smokers, joint upregulation in miR-320c, miR-200c-3p, and miR-449c-5p levels in the miRNA profiling of BAL samples is detected. These findings might suggest that 3-miRNA signature might be potentially used as biomarkers useful to distinguish COPD patients from smokers and non-smoker subjects [79].

The high-throughput RNA sequencing describes a differential expression of 282 mRNAs, 146 lncRNAs, 85 miRNAs, and 81 circRNAs in peripheral blood of COPD patients compared with control. GSEA analysis shows that these differentially expressed RNAs correlate with several critical biological processes such as “ncRNA metabolic process”, “ncRNA processing”, “ribosome biogenesis”, “rRNAs metabolic process”, “tRNA metabolic process”, and “tRNA processing”. All of them might participate in the progression of COPD. These data determine the construction of the lncRNA-mRNA co-expression network, and the constructed circRNA-miRNA-mRNA in COPD opens new perspective in the nc-RNA involvement as potential regulatory roles in COPD [80]. lncRNA-proliferation, apoptosis, inflammation, migration, and epithelial-mesenchymal transition (EMT) are cell processes controlled by miRNA-mRNA network. Many lncRNA-miRNA-mRNA are biomarker indicators of comorbidities and may be considered as therapeutic targets for chronic inflammatory diseases of the airways of both COPD and asthma [81]. Numerous biological processes are due to the irreversible molecular changes caused by cigarette smoking in COPD patients. Several studies show its direct correlation with the dysregulation of different miRNAs suggesting the diagnostic/prognostic potential of miRNA-based biomarkers and their efficacy as therapeutic targets [82] (Figure 3).

In conclusion a relevant number of recent studies support the concept about the potential role of miRNAs network in the regulation of different cellular processes, such as proliferation, apoptosis, inflammation, migration, and EMT in COPD patients. These data support the concept of their biological activities in the relevant pathophysiological processes of chronic inflammatory airway diseases. In this scenario, we comprehensively underline the miRNA network activities in different cell types and their potential roles as biomarkers, indicators of comorbidities, or therapeutic targets for COPD (Figure 3).

## 5. ncRNAs in Lung Cancer

COPD and epigenetic events are involved in the development of LC [83]. miRNAs (miR-21, miR-200b, miR-210, and miR-let7c) and DNA methylation have higher levels in patients with LC showing a history of COPD than in patients with LC alone. Often, patients with LC show a declared history of an underlying respiratory disease. The identification of airway diseases in all patients with LC can represent a differential biological profile, involved in the determination of tumor progression and therapeutic response. In these patients, biomarkers of mechanisms involved in tumor growth, angiogenesis, migration, and apoptosis are differentially expressed in tumors of patients with underlying respiratory disease. Additionally, epigenetic events offer a niche for pharmacological therapeutic targets [83].

Several miRNAs are linked to both inflammatory and proliferative processes both observed in inflammatory and cancer diseases of the lung. For instance, miR-21 had a role in both inflammation and cancer and is linked to cigarette smoking-related conditions of both, patients with COPD and LC. MiR-21 is downregulated in skeletal muscle of patients with COPD patients than in non-smoking controls and its levels of expression well correlated with clinical features [83]. MiR-21 are highlighted for its critical role in LC, such as adenocarcinoma, non-small cell lung cancer. Accordingly, miR-21 is involved in various cell processes including cell death to cancer stemness. The expression of miR-21 is higher in biofluids and tissues of cancer, representing valuable potential markers of diagnosis and prognosis of LC [84]. Likewise, epidermal growth factor receptor (EGFR)-mutated lung had considerably increased miR-21 expression compared to those without mutations. EGFR can affect miRNA maturation by posttranslational modification of AGO2 highlighting the relevant relationship between a LC mutation and miR-21 status [85]. MiR-21 is commonly overexpressed in LC, where mutations are strongly related with miR-21 and its target proteins. Non-small-cell lung cancer (NSCLC) is consistent of R175H- and R248Q-mutant p53, and miR-21 is upregulated. Patients with elevated expression of p53 mutations and higher levels of miR-21 had a lower overall survival rate [86].

miRNA-155 is a marker of early diagnosis and monitoring of cancer diseases. It is highly expressed in tumor cells of LC. An electrochemical sensor propelled by exonuclease III, which is coupled with multiple signal amplification strategies for highly efficient microRNA-155 detection with a limit of detection of 0.035 fM. Overall, the strategy for miRNA detection offers good prospects for early cancer screening [87]. miR-942 is indicated as a prognostic marker for early discovery of tumor progression, metastasis, and development. Dysregulation of miR-942 amounts is identified in patients with non-small-cell LC, and they indicate as biomarkers in cancer discovery and assist in therapy control due to their epigenetic involvement in gene expression and other biological cell processes. In this manner, due to its involvement in cell proliferation, migration, and invasion through cell cycle pathways, miRNA-942 is considered as a potential candidate for prediction of LC [88].

miR-320a-3p, miR-210-3p, miR-92a-3p, miR-21-5p, and miR-140-3p are indicated in the literature with a predictive performance in the identification and in the pre-diagnostic setting of LC cases. They are increased in the serum of patients with LC in comparison with control subjects. miR-320a-3p, miR-210-3p, miR-92a-3p, miR-21-5p, and miR-140-3p compared to of surfactant protein B (Pro-SFTPB), cancer antigen 125 (CA125), carcinoembryonic antigen (CEA), and cytokeratin-19 fragment (CYFRA21-1) precursor forms improved sensitivity at statistical analysis to detect the condition of LC diseases. These data demonstrate that miRNAs in combination with a panel of proteins might be considered as useful tools for an early detection of LC [89] (Figure 3).

## 6. Environmental Pollution and ncRNAs in the Airways

Among the emerging biomarkers associated with the effects of environmental contaminants in the respiratory system, there are markers of oxidative stress (ROS, MDA, GSH, etc.,), inflammation (interleukins, PHENO, CC16, etc.), DNA damage (8-OHdG, γH2AX, OGG1). In addition to these biomarkers, the action of pollutants on respiratory system is indicated to indicators of epigenetic modulation (DNA methylation, histone modification, miRNA) playing a fundamental role. However, studies that investigate miRNA expressions and functions in lung diseases association with air pollution are scarce. Interventions in public health requires the detection of specific biomarkers to define PM_2.5_-elicited inflammation, fibrogenesis, and carcinogenesis. Some inconsistent findings may possibly relate to the inter-study differentials in the airborne PM_2.5_ sample, exposure mode, and targeted subjects, as well as methodological issues. The identification of novel, specific biomarkers by a scientific approach obtained with omic-techniques might be useful to define the causal relationship between PM_2.5_ pollution and deleterious lung outcomes by [90].

Exposure to airborne fine particulate matter as PM2.5 has short- and long-term adverse effects on lung functions. However, early impairment of lung function is not easily detectable in time. In particular, miRNAs are classified as novel biomarkers for PM-related injury in lung diseases, and currently are widely used in epidemiological and toxicological studies to understand the biological mechanisms underlying the adverse health outcomes of PM2.5 [91]. MiR-146a and miR-146b are elevated remarkably in bronchoalveolar lavage fluid (BALF) and lung tissue homogenate of BALB/c mice exposed to PM2.5. These data suggest the relationship between MiR-146a and miR-146b and pulmonary dysfunction after the exposure to the toxicants [92]. MiR-217-5p suppresses inflammation, oxidative stress, and lung injury in macrophages and lung tissue in a mouse model, showing activated STAT1-signal after the exposure to PM2.5 [93]. The trigger with PM_2.5_ significantly enhances the biological behaviors of A549 cells promoting EMT transformation. The knockdown of miR-582-3p changes the effects of PM_2.5_ on malignant biological behavior in A549 cells reducing Wnt/β-catenin signaling pathway and EMT. These data suggest that the over expression of miR-582-3p after the exposure to PM_2.5_ in the environment might be involved in the mechanisms of LC [94]. PM_2.5_ and related epigenetic modifications are involved in asthma pathogenesis; however, the mechanism remains unclear. The exposure to traffic-related PM_2.5_ aggravated pulmonary inflammation in rats and increased the level of miR146a while decreased the level of miR-31. These epigenetic modifications provide a new target for asthma treatment and control, associated with their negative action on the regulatory T (Treg) cells function and T-helper type 1 (Th1)/Th2 cells imbalance causing exacerbation of inflammation [95]. Increased levels of miR-155 in the serum of asthmatic children correlate with particulate matter level exposure. Recently, it is reported that miRNA post-transcriptional regulation that involves RNA-based epigenetic mechanisms represents a key epigenetic factor of asthma pathogenesis associated with air pollution [96].

The use of miRNAs as biomarkers and as preventive targets for childhood asthma represent an attractive RNA hypothesis. In fact, children with severe bronchiolitis exposed to higher levels of air pollution show higher risk of developing asthma than children exposed to lower levels [97]. Air pollution aggravate type 2 responses, and lead to an increase in neutrophils as a source of miRNAs in the airway [96]. However, studies on adult asthma identify that numerous miRNAs may be involved in a better identification and understanding of the effect of environmental pollution in airway disease. In bronchial brushing, the exposure of atopic individuals to diesel exhaust and allergen shows miR-183, -324, and -132 expression modulated by allergen but not by diesel exhaust [98]. Moreover, diesel exhaust exposure increases expression of miR-21, miR-30e, miR-215, and miR-144 in the plasma of mild asthmatics enrolled in a randomized crossover study. Importantly, miR-21 and miR-144 expression is associated with increased oxidative stress markers and with a reduced antioxidant gene expression [99].

An increasing body of studies has focused on the effect of PM_2.5_ on lung adenocarcinoma; however, also in this case the mechanism remains unclear. It is described that the exposure in patient-derived xenograft (PDX) models to PM2.5 can generate tumorigenesis and metastasis in lung adenocarcinoma of patient-derived xenograft (PDX) models, and migration and invasion in lung adenocarcinoma cell lines. PM2.5 are involved in the regulation of miRNAs including miR-30A, miR-125A, miR-200A, miR-200C, miR-221, and Let-7c of cancer stem cells (CSCs) pathway in lung adenocarcinoma cells [100] (Figure 3).

## 7. ncRNAs as Therapeutic Approach in Lung Diseases

Inflammatory diseases of the airways represent a relevant problem for lung health often related to the activation of molecular mechanisms. For this reason, it is necessary to define new pharmacological approaches to overcome the effectiveness of existing conventional therapeutic therapies and to address fundamental issues concerning specific molecular pathways [101]. The efficacy of therapy targeting pro-inflammatory miRNAs in mouse models of mild/moderate and severe asthma is recently established [102]. The suggested approaches are principally directed toward miRNAs and antagonists that mimics or blocks the specific activities to be used in vivo. However, their use in the local tissue of the lung might represent a limit for their use as therapeutic treatment limiting the adequate clinical applications. The knowledge of chemically nature of miRNAs is useful to understand the stability of themselves miRNA or antagomir in the blood, cell permeability, and optimized its target specificity. The knowledge of the miRNA’s nature might guide an adequate lung cell uptake, high target specificity, and efficacy with tolerable off-target effects. Innovative approaches to enhance RNA stability, tissue targeting, cell penetration, and intracellular endosomal escape are critical to realize the full potentials of RNA drugs.

It is observed that circulating miRNA would reveal candidate biomarkers related to airway hyperresponsiveness (AHR) and provide biologic insights into asthma epigenetic influences. Eight serum miRNAs, including miR-296–5p, are associated with PC20 in the Childhood Asthma Management Program (CAMP) cohort [103]. In ovalbumin (OVA)-induced asthma model established in female BALB/c mice, it is observed that targeting miRNA-182-5p is a possible new strategy to treat asthma. In fact, the treatment of female BALB/c mice with miRNA-182-5p agomir significantly reduces the levels of IL-4, IL-5, OVA-induced IL-13, and eosinophil percentage in bronchoalveolar lavage fluid, including Th2 inflammatory factors downregulation. Furthermore, miRNA-182-5p agomir reduces the peribronchial inflammatory cell infiltration, goblet cell proliferation, and collagen deposition [104]. To help with the concept of personalized medicine, it is necessary to identify novel biomarkers involved in the disease. The objective is to improve the knowledge of disease phenotype and its classification, to better identify the pharmacological treatment. MiR-144-3p is increased in both lungs and serum of asthma patients. It was observed that the levels of miR-144-3p in the lung correlated with blood eosinophilia and with the expression of genes, strongly related to the pathophysiology of asthma, while the levels of miR144-3p in serum is associated with higher doses of corticosteroids in severe asthmatic patients. These data suggest that miR-144-3p reaches higher levels in severe diseases in association to corticosteroid treatment [105].

MiRNAs operate as posttranscriptional regulators, providing another level of control of GC receptor levels. Many actual experimental data describe miRNAs as useful biomarkers providing a promising approach to better characterize and treat patients with airway diseases [106]. miR-21 play a significant role in the pathogenesis of asthma and in steroid-insensitive experimental asthma steroid resistance via PI3K activation. These findings propose that the development of miRNA-based drugs could constitute a promising therapy to improve treatment of GC-resistant asthma by amplifying phosphoinositide 3-kinase-mediated suppression of histone deacetylase 2 [107].

The emerging attention on ncRNAs in airway disease is focused on the use of siRNAs as regulatory ncRNAs [108]. The therapeutic effects of synthetic siRNA are demonstrated in allergen-induced asthma models [29]. These data suggest improving and to deepen the knowledge of the role of the ncRNAs, to describe a new direction in the field of targeted asthma in adult and children therapy and in COPD [29]. In asthma and COPD GC resistance is controlled by many molecular and cellular pathobiological mechanisms. Actually, patients with GC resistance are treated with broad-spectrum anti-inflammatory drugs that often have major side effects [106]. Recently, the effect of ncRNAs on asthma and COPD attracted the attention of researchers as a new molecular mechanism to target, with the aim to contribute a better treatment of inflammatory airway diseases. Studies on this field are lacking though. MiRNAs are non-coding molecules that act both as regulators of the epigenetic landscape and as biomarkers for diseases, including asthma and COPD. Numerous ncRNAs such as miRNAs, lncRNAs, and circRNAs, are linked to COPD, but today only few nc-RNAs are functionally characterized. The use of nc-RNA including miR-195, miR-181c, and TUG1 as therapeutic targets might be considered as promising in the control of COPD in vivo. The development of innovative drugs includes siRNA therapeutics targeting mRNAs critical for the pathogenesis of COPD. For instance, siRNAs targeting RIP2, RPS3, MAP3K19, and CHST3 mRNAs are successfully validated in an in vivo COPD model. Furthermore, it is described that the best route for the administration of RNA therapeutics in the lungs of COPD patients is inhalation [109]. TLR2/4 signaling are controlled by miR-27-3p expression, a nc-RNA involved in the production of pro-inflammatory cytokines through targeting the 3^/^-UTR sequences of ppARγ, suppressing ppARγ activation and miR-27-3p in alveolar macrophages (AMs). These data provide the information that miR-27-3p might be considered as a therapeutic method able to control the airway inflammation in COPD patients [110].

Fibroblasts from the lung of COPD patients show changes associated with an altered production of growth factors, fibronectin, and inflammatory cytokines [111]. For instance, vascular endothelial growth factors (VEGF) contribute to disturb vasculature in the lung. It is observed that the levels of miR-503 expression are lower in fibroblasts compared to the lung of a COPD [112] patient and show a positive correlation with increased levels of VEGF release. These data suggest that miR-503 production is involved in the control vascular homeostasis in COPD and helps to consider this miRNA as a therapeutic target in COPD [113,114]. DNA-based gene therapy has a reversible alternative with RNA therapeutics that is highly specific and safer. Fomivirsen, mipomersen, defibrotide, eteplirsen, nusinersen, inotersen, and patisiran are seven oligonucleotide-based drugs approved for a variety of disease conditions. These RNA drugs along with many others are candidate to be tested in clinical trials [115] and might also work in COPD [109].

Treatment of NSCL lung cancer is conditioned by NSCLC cell resistance to cisplatin. This topic represents a very important therapeutic challenge. The lnc-miRNA LINC02389 regulates cell proliferation and promoted cell apoptosis in NSCLC. Cisplatin-resistant cells is guided by an overexpression of oxidative stress biomarkers and regulated by LINC02389. The lnc-miRNA is highly expressed in NSCLC tissues and is associated with poor prognosis of NSCLC patients. Cisplatin-resistant NSCLC cells shows LINC02389 overexpressed, while miR-7-5p is downregulated. In these cells, LINC02389 negatively correlates with the expression of miR-7-5p. This last exerts an opposite effect and is as spongin for LINC02389 in NSCLC. These data might suggest therapeutic solution by regulating the expression of miR-7-5p in cisplatin resistance in NSCLC [116].

MiRNAs can be explored in early diagnosis and treatment strategies to prevent LC. Many of them have a relevant role in the specific cell cycle core regulation. These observations indicate the need to provide information with the aim to create new perspectives on cell-cycle-associated miRNA studies as target and therapeutics in LC treatment. This last consideration is well supported in the review of Fariha et al. [117]. The need to create more and more new perspectives are dictated by the importance of miRNAs in the pathogenesis of LC. For instance, miR-10b, miR-21, miR-150, miR-222, miR-96, miR-1290, miR-499 are miRNAs overexpressed in LC. Many authors suggest the use of antagomir, better known as anti-miRNAs, to block their negative action in LC. In fact, specific antagomirs are studied to contrast the activities of miR-10b, miR-21, miR-150, miR-222, miR-96, miR-1290, miR-499 blocking the interaction between the RISC complex and the target mRNA, thereby preventing mRNA translation [118].

## 8. Extracellular Vesicles and nc-RNA in Airway Diseases

Highest vascular density is a characteristic of the lung. In the lung cells as macrophages, fibroblasts, epithelial cells, and endothelium are involved in the circulation of extracellular vesicles (EVs), including exosomes, microvesicles, and apoptotic bodies [119]. Exosomes are small vesicles with a lipidic nature, deputies to the transport of proteins, lipids, and RNA molecules. Their immunological function is facilitated cell-to-cell communication under normal and diseased conditions. miRNAs and proteins present in EVs are ideal non-invasive predictive useful tools to contribute to an early diagnosis, prognosis, and therapeutic targets in lung disease, since they are factors associated with important information on biological responses. EVs released from various cells serve as mediators of information exchange between different cells to regulate a more accurate molecular mechanism involved in the process of cell-to-cell communication. MiRNAs are shuttled by EVs playing a pivotal role in the pathogenesis of respiratory diseases. EV miRNAs show promise as diagnostic biomarkers and therapeutic targets in several lung diseases [120] (Figure 4).

Nc-RNAs play critical roles in physiological and pathological processes of LC. EVs contain nc-RNAs packaged and are transported between LC cells and stromal cells. In this manner, EVs can regulate multiple activities of malignant cells of LC such as proliferation, migration, invasion, epithelial-mesenchymal transition, metastasis, and treatment resistance. In fact, it is possible to detect EVs in various body fluids associated with the stage, grade, and metastasis of LC, and potentially serve as diagnostic and prognostic biomarkers of disease playing a pivotal role in the clinical treatment of LC [122]. miR-153-3p-EVs are involved in damaging respiratory functions and produce a mass of inflammatory cells around the lung tissue of mice. It is observed that antagomir-153-3p treatment controls the deterioration of respiratory functions and inhibits the growth of lung tumors in mice. This study suggests the potential molecular mechanism of miR-153-3p-EVs in the development of metastasis of adenocarcinoma and provides a potential strategy for the treatment of metastasis in the lung [123].

MiRNA EVs cargo is different between patients with small-cell lung cancer (SCLC) and NSCLC. Particularly, miR-331-5p, miR-451a, miR-363-3p can distinguish SCLC and NSCLC tumor with highest rates of specificity and sensitivity [124]. Furthermore, EVs cargo with 7 miRNAs (miR-451a, miR-486-5p, miR-363-3p, miR-660-5p, miR-15b-5p, miR-25-3p, and miR-16-2-3p) differentiate NSCLC patients and healthy subjects [125].

Let-7i-5p is significantly overexpressed in PM_2.5_-EVs and asthmatic plasma; and its levels of expression well correlated with PM_2.5_ exposure in children with asthma. Mechanistically, let-7i-5p is packaged into PM_2.5_-EVs by interacting with ELAVL1 and internalized by both “horizontal” recipient HBE cells and “longitudinal” recipient-sensitive HBSMCs. The result is the activation of MAPK signaling pathway via suppression of DUSP1 as its target. Furthermore, an injection of EV-packaged let-7i-5p into PM_2.5_-treated juvenile mice aggravated asthma symptoms. The conclusion is that PM generates childhood asthma attacks via extracellular vesicle-packaged let-7i-5p-mediated modulation of MAPK pathway [126].

## 9. Conclusions

Lung diseases are a cause of morbidity and mortality in the world for all age groups. However, the underlying molecular mechanisms involved in airway diseases are not fully explored; overall, those associated with the epigenetic modification of ncRNAs differentially expressed in diverse samples of tissue and blood. We reported here some data on the role of ncRNAs in lung disease, to underline the most recent knowledge regarding their biological and molecular functions. They might be known for their capability of being biomarkers or for having a specific role in the pathogenesis of lung diseases.

ncRNA have recently attracted much attention for their roles in the regulation of a variety of biological processes. Today few biomarkers and drugs targeting ncRNAs have been identified as potential tools for clinical diagnosis and treatment, so that it becomes more and more important to verify the applicability of ncRNAs in clinical management of lung diseases. Researchers should combine innovative data-driven model elaboration and model-driven experimental design to elucidate how all ncRNAs cooperate in the pathogenesis and diagnosis of lung diseases. Finally, it goes without saying the need to develop a new approach aimed at the combination of bioinformatics, basic immunology, RNA biology, genomics, and proteomics, to better understand the role of ncRNAs regulatory networks in the pathogenesis of lung diseases. The final goal must be making a new direction in the diagnostics and pharmacological therapeutics for the lung.

## Figures and Tables

**Figure 1 cancers-15-00054-f001:**
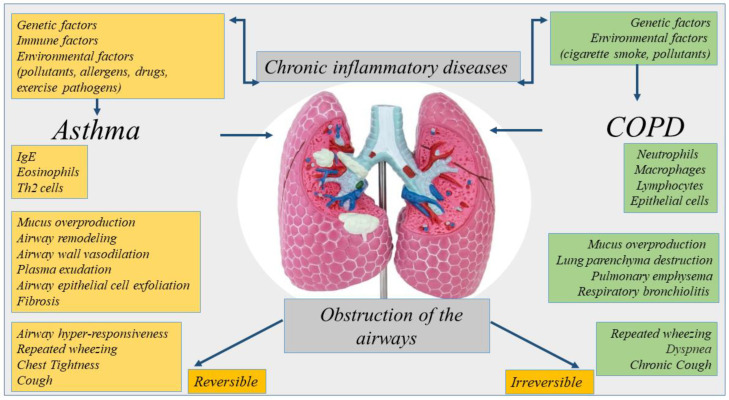
Representation of the etiology, inflammation, and symptoms of chronic inflammatory diseases of the lung. Asthma and COPD are diseases with similarities. However, they are different regarding etiology, type of inflammatory cells, mediators, consequences of inflammation.

**Figure 2 cancers-15-00054-f002:**
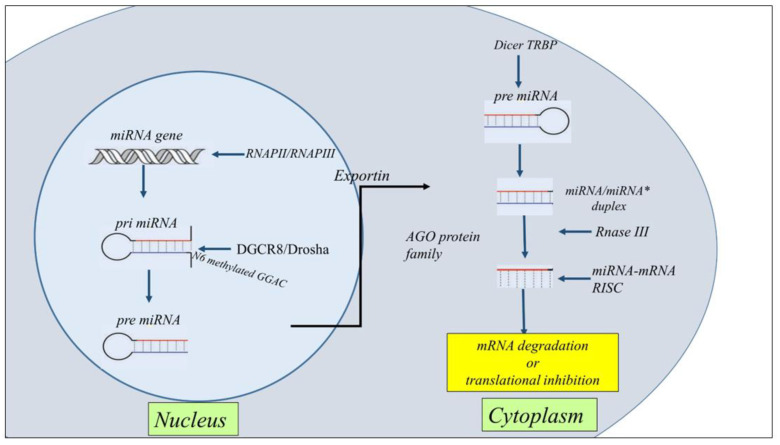
Schematic representation of the general steps of miRNA biogenesis. The biogenesis of miRNAs starts in the nucleus with the processing of RNA polymerase II/III transcripts post- or co-transcriptionally into pri-miRNAs. Then, a ribonuclease III enzyme transforms the pri-miRNAs in pre-miRNAs, which are transported into the cytoplasm by Exportin 5 (XPO5)/RanGTP complex action. Reaching the cytoplasm pre-miRNA is transformed in mature miRNA/miRNA* duplex by RNase III endonuclease Dicer and the RNA-binding protein TRBP the action. miRNA then forms a miRNA-mRNA RISC complex to induce mRNA degradation or translational repression.

**Figure 3 cancers-15-00054-f003:**
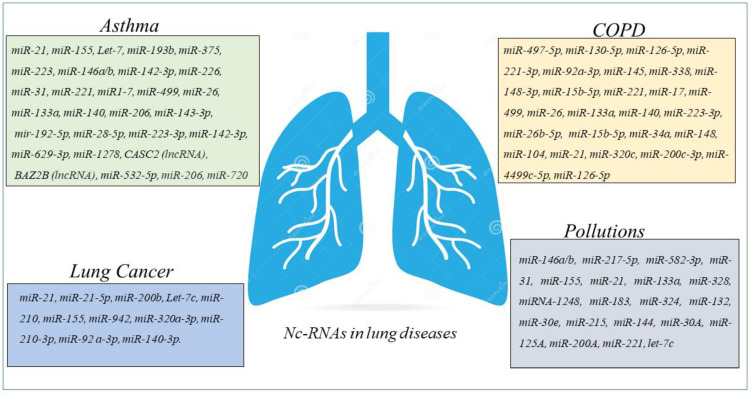
Schematic panel describing nc-RNAs involved in airway diseases and indicated in the review. The miRNAs and lnc-RNAs are expressed in lung diseases such as asthma, COPD, LC, or in lung diseases associated with environmental pollution.

**Figure 4 cancers-15-00054-f004:**
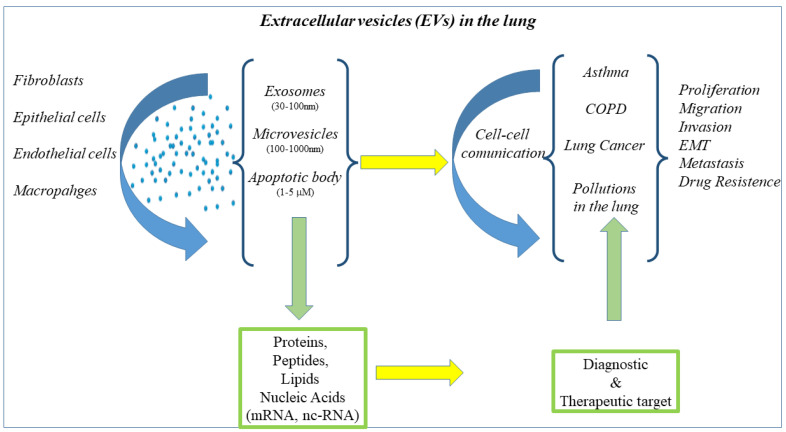
Cells of the lung release EVs of different size containing proteins, lipids, and nucleic acid involved in the mechanisms of cell–cell communication underlies the immunological response of respiratory disease such as asthma, COPD, LC, and airway diseases related to pollutions. EVs content show promise as diagnostic biomarkers and therapeutic targets in several lung diseases. BALF- and lung-tissue-derived EVs of healthy non-smokers, smokers have a miRNA profile with three differently expressed miRNAs in BALF, and one in the lung-derived EVs from COPD patients as compared to healthy non-smokers. MiR-122-5p is three- or five-fold downregulated among the lung-tissue-derived EVs of COPD patients as compared to healthy non-smokers and smokers, respectively. These data strongly suggest that miRNAs in the lungs of subjects with chronic lung diseases might be considered as potential biomarkers useful for therapeutic targets [121].

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
