# Peer review of "Non-Coding RNAs in Airway Diseases: A Brief Overview of Recent Data"

_cancers, 2022, doi:10.3390/cancers15010054_

Round 1

Reviewer 1 Report

Albano et al. summarized the main ncRNAs involved in Asthma, COPD and Lung cancer. It is a very interesting review that gives a complete state-of-art of the involvement of microRNAs in the pathogenesis of lung diseases. 

The authors could add a figure that may summarize the paragraph: "Extracellular vesicles and ncRNA in airway diseases".

Author Response

Reviewer 1

General comments and suggestions for Authors: Albano et al. summarized the main ncRNAs involved in Asthma, COPD and Lung cancer. It is a very interesting review that gives a complete state-of-art of the involvement of microRNAs in the pathogenesis of lung diseases. 

Response to the general comment: We thank the reviewer for the kind general comment about the contents of our review.

Comment #1: The authors could add a figure that may summarize the paragraph: "Extracellular vesicles and ncRNA in airway diseases".

Response to comment #1: we thank the reviewer for the suggestion and accordingly we added to the revised manuscript a figure that summarize the paragraph: "Extracellular vesicles and ncRNA in airway diseases". Please see page 14 of the revised manuscript

Reviewer 2 Report

This is a nicely written review. Here are some minor comments for the authors' consideration:

1. Line 109: it should be three noteworthy ncRNAs (miRNAs, lncRNAs, circRNAs).

2. Line 115: will be good to include the role(s) of circRNAs?

3. Line 154: consistency of nomenclature. ncRNAs in Asthma. Also, we know that asthma can be further classified as mild, moderate and severe. The authors may want to discuss the roles of miRNAs and ncRNAs in asthma subtyping. These ncRNA biomarkers may help to triage asthma patients. Are these biomarkers found in sputum supernatant too (this is briefly mentioned in the COPD section but are there more examples?) and what is the advantage of using serum as the input material for ncRNA biomarker detection?

4. Typo error in Figure 3. Pollution. Also, check consistency of miRNA nomenclature (ncRNAs in lung diseases too). 

5. Line 437: check consistency of miRNA nomenclature.

6. Line 449-451: can the authors rephrase this sentence "To know...." It makes little sense to me at the present form. In fact, there are lots of grammatical errors in the entire section 7 (ncRNAs as therapeutic approach in lung diseases); please revise this section accordingly.

Author Response

Reviewer 2

General Comments and Suggestions for Authors: This is a nicely written review. Here are some minor comments for the authors' consideration:

Response to the general comment: We thank the reviewer for the kind general comment about the contents of our review. Accordingly, we tried to answer to your comments.

Comment #1. Line 109: it should be three noteworthy ncRNAs (miRNAs, lncRNAs, circRNAs).

Response to comment #1: We thank the reviewer for the observation. As described in the abstract section “In this review we describe several of most recent knowledge concerning ncRNA (overall miRNAs) expression and activities in the lung”. Please see the page 1, line 27 to 30.

However, as you had suggested, we described  miRNAs, lncRNAs, circRNAs in the section biogenesis of nc-RNA. Please see the nc-RNA biogenesis section, page 3 – 4, line 129 to 147 of the revised version of the manuscript.

Comment 2#. Line 115: will be good to include the role(s) of circRNAs?

Response to comment #2: We now enclosed the description of the role of circRNAs. Please see the nc-RNA biogenesis section, page 3 – 4, line 134 to 147 of the revised version of the manuscript.

Comment 3#. Line 154: consistency of nomenclature. ncRNAs in Asthma. Also, we know that asthma can be further classified as mild, moderate, and severe. The authors may want to discuss the roles of miRNAs and ncRNAs in asthma subtyping. These ncRNA biomarkers may help to triage asthma patients. Are these biomarkers found in sputum supernatant too (this is briefly mentioned in the COPD section but are there more examples?) and what is the advantage of using serum as the input material for ncRNA biomarker detection?

Response to comment #3: we thank the reviewer for the suggestions. Accordingly we insert some adequate sentences in the revised version of the manuscript. Please see page 5, line 209-215.

We did not enclose the information about the detection of microRNa in the serum since in particular we focused our attention on miRNa and not circRNAs

Comment 4#. Typo error in Figure 3. Pollution. Also, check consistency of miRNA nomenclature (ncRNAs in lung diseases too). 

Response to comment #4: Accordingly, we checked

Comment 5#:  Line 437: check consistency of miRNA nomenclature.

Response to comment #5: Accordingly, we checked

Comment 6#: Line 449-451: can the authors rephrase this sentence "To know...." It makes little sense to me at the present form. In fact, there are lots of grammatical errors in the entire section 7 (ncRNAs as therapeutic approach in lung diseases); please revise this section accordingly.

Response to comment #6: Many thanks for the suggestions. Accordingly, we rephased the sentences., We hope that now the sense is clear, please see page 11 line 507- 511.
